# ON BATCH ADAPTIVE TRAINING FOR DEEP LEARNING: LOWER LOSS AND LARGER STEP SIZE

## ABSTRACT

Mini-batch gradient descent and its variants are commonly used in deep learning. The principle of mini-batch gradient descent is to use noisy gradient calculated on a batch to estimate the real gradient, thus balancing the computation cost per iteration and the uncertainty of noisy gradient. However, its batch size is a fixed hyper parameter requiring manual setting before training the neural network. Yin et al. (2017) proposed a batch adaptive stochastic gradient descent (BA-SGD) that can dynamically choose a proper batch size as learning proceeds. We extend the BA-SGD to momentum algorithm, and evaluate both the BA-SGD and the batch adaptive momentum (BA-Momentum) on two deep learning tasks from natural language processing to image classification. Experiments confirm that batch adaptive methods can achieve a lower loss compared with mini-batch methods and methods that manually increase the batch size, after scanning the same epochs of data. Furthermore, our BA-Momentum is more robust against larger step sizes, in that it can dynamically enlarge the batch size to reduce the larger uncertainty brought by larger step sizes. We also identified an interesting phenomenon, *batch size boom*. The code implementing the batch adaptive framework is now open source, applicable to any gradient-based optimization problems.

## 1 INTRODUCTION

Efficiency of training large neural networks becomes increasingly important as deep neural networks tend to have more parameters and require more training data to achieve the state-of-the-art performance on a wide variety of tasks (Goodfellow et al., 2015). For training deep neural networks, stochastic gradient descent (SGD) (Robbins & Monro, 1951) and its variants, including momentum, which utilizes past updates with an exponential decay (Qian, 1999), and other methods that can adapt different learning rates for each dimension, such as ADAGRAD (Duchi et al., 2010), ADADELTA (Zeiler, 2012) and ADAM (Kingma & Ba, 2014), are commonly used.

SGD approximates the gradient by only using a single data instance in each iteration, which may lead to uncertainty of approximation. This uncertainty can be reduced by adopting a batch of instances to do the approximation. In mini-batch SGD, the batch size is a fixed hyper parameter requiring manual setting before training the neural network. Setting the batch size typically involves a tuning procedure in which the best batch size is chosen by a series of attempts.

Yin et al. (2017) has developed a batch adaptive stochastic gradient descent (BA-SGD) that can dynamically choose a proper batch size as learning proceeds. BA-SGD models the decrease of objective value as a Gaussian random walk game with rebound on the basis of Taylor extension and central limit theorem. Its core idea is to only update the parameters when the ratio between the expected decrease of objective value and the current batch size is large enough, otherwise enlarge the batch size to better approximate the gradient. It claimed that by smartly choosing the batch size, the BA-SGD not only conserves the fast convergence of SGD algorithm but also avoids too frequent model updates, and compared with mini-batch SGD, its objective value decreases more, after scanning the same amount of data.

However, the experiment in Yin et al. (2017) was only conducted on some simple classification tasks using fully connected neural network with one input layer, one output layer and two hidden layers. What about the evaluation on some complex neural networks, such as convolutional neural network (CNN) and recurrent neural network (RNN)? How well would the batch adaptive algorithm

perform on other complicated tasks related to natural language processing and computer vision? Furthermore, empirical studies reveal that SGD usually performs not so well in some deep and complex neural networks (Sutskever et al., 2013). Can this batch adaptive framework be extended to other gradient based optimization algorithms except SGD?

Therefore, in this paper we extend the batch adaptive framework to momentum algorithm, and evaluate both the batch adaptive SGD (BA-SGD) and the batch adaptive momentum (BA-Momentum) on two deep learning tasks from natural language processing to image classification. These two tasks use RNN and CNN respectively, which cover most of the deep learning models.

In our experiments, we have the following observations. First, for batch adaptive methods, their loss functions converge to lower values after scanning same epoches of data, compared with fixed-batch-size methods. Second, BA-Momentum is more robust against large step sizes by dynamically enlarging the batch size to counteract with the larger noise brought by larger step sizes. Third, we observed a *batch size boom*, a concentrated period where the batch size frequently increases to larger values, in the training of BA-Momentum. The batch size boom is of significance in that it always appears at the point where mini-batch method starts to reach its lowest possible loss and it helps BA-Momentum keep dropping to even lower loss. More details on these observations and their analysis can be found in Section 4. The code implementing the batch adaptive framework using Theano (AlR) is now open source [1], which is applicable to any gradient-based optimization problems.

This paper is organized as follows. In Section 2, we briefly introduce the batch adaptive framework proposed by Yin et al. (2017). In Section 3, we extend the batch adaptive framework to momentum algorithm. In Section 4, we demonstrate the performance of BA-M and BA-SGD on Fashion-MNIST (Xiao et al., 2017) and relation extraction task, and then reveal the robustness of BA-Momentum against large step sizes. In Section 5, we discuss some efficiency issue concerned with implementing this batch adaptive framework, and also propose several promising applications based on this framework.

## 2 PREREQUISITES

In this section we briefly summarize the batch adaptive stochastic gradient descent proposed by Yin et al. (2017).

### 2.1 NOTATIONS

We use $\mathcal{X}$ and $\mathcal{Y}$ to respectively denote the training data set and its random subset. The vector of model parameters is denoted by $\vec{\theta}$ and is subscripted by $t$ to denote an iteration. F is used to denote objective function, while $\mathtt{f}_{\vec{\theta}}$ is the partial derivative of function F with model parameters $\vec{\theta}$ (i.e. the gradient, computed over the whole data set).

### 2.2 BATCH ADAPTIVE FRAMEWORK

Let $\vec{\xi}_j$ denote the difference between the approximate gradient computed on an individual instance $\mathtt{g}_{\vec{\theta}}^j$ and the real gradient $\mathtt{f}_{\vec{\theta}}$, thus, $\hat{\mathtt{f}}_{\vec{\theta}}$, the approximate gradient computed on a batch $\mathcal{Y}$ can be written as:

$$\hat{\mathtt{f}}_{\vec{\theta}} = \frac{\sum_{\mathbf{y}_j \in \mathcal{Y}} \mathtt{g}_{\vec{\theta}}^j}{|\mathcal{Y}|} = \mathtt{f}_{\vec{\theta}} + \frac{\sum_{\mathbf{y}_j \in \mathcal{Y}} \vec{\xi}_j}{|\mathcal{Y}|} \tag{1}$$

The term $\frac{\sum_{\mathbf{y}_j \in \mathcal{Y}} \vec{\xi}_j}{|\mathcal{Y}|}$ can be viewed as a random variable determined by the randomly sampled batch $\mathcal{Y}$. On the basis of Central Limit Theorem (CLT), it should satisfy a multi-dimension normal distribution $\mathcal{N}(\mathbf{0}, \frac{\Sigma}{|\mathcal{Y}|})$, where $\Sigma$ is the covariance matrix of $\vec{\xi}_j$. $\mathtt{f}_{\vec{\theta}}$ is the real gradient, computed on the whole data set and treated as a constant given the parameters $\vec{\theta}$. Thus we have $\hat{\mathtt{f}}_{\vec{\theta}} \sim \mathcal{N}(\mathtt{f}_{\vec{\theta}}, \frac{\Sigma}{|\mathcal{Y}|})$.

---

[1] The code implementing the batch adaptive framework using Theano can be found in our GitHub repository: https://github.com/thomasyao3096/Batch_Adaptive_Framework

After modelling the estimation of gradient on a batch as a normally distributed random variable, Yin et al. (2017) uses first-order Taylor extension to approximate the objective function $\mathbf{F}(\vec{\theta})$ at any parameter configuration. The equation is shown below.

$$\mathbf{F}(\vec{\theta}) = \mathbf{F}(\vec{\theta}_0) + \mathbf{f}_{\vec{\theta}_0}^T \cdot (\vec{\theta} - \vec{\theta}_0) + \mathbf{h}_{\vec{\theta}_0}(\vec{\theta}) \tag{2}$$

where the function $\mathbf{h}_{\vec{\theta}_0}(\vec{\theta})$ is the remainder term satisfying $\lim_{\vec{\theta} \to \vec{\theta}_0} \mathbf{h}_{\vec{\theta}_0}(\vec{\theta}) = 0$.

If SGD optimization algorithm is adopted to update the parameters, with Equation (1) and Equation (2), the decrease of objective value can be expressed as:

$$\Delta \mathbf{F}(\vec{\theta}_0) = \mathbf{F}(\vec{\theta}_0 - \eta \hat{\mathbf{f}}_{\vec{\theta}_0}) - \mathbf{F}(\vec{\theta}_0) \approx -\eta \mathbf{f}_{\vec{\theta}_0}^T \cdot (\mathbf{f}_{\vec{\theta}_0} + \frac{\sum_{\mathbf{y}_j \in \mathcal{Y}} \vec{\xi}_j}{|\mathcal{Y}|}) = -\eta \cdot \mathbf{f}_{\vec{\theta}_0}^T \mathbf{f}_{\vec{\theta}_0} - \eta \cdot \mathbf{f}_{\vec{\theta}_0}^T \cdot \vec{\varepsilon}_{\mathcal{Y}} \tag{3}$$

where $\eta$ is the learning rate, and the noise term satisfies $\vec{\varepsilon}_{\mathcal{Y}} \sim \mathcal{N}(\mathbf{0}, \frac{\Sigma}{|\mathcal{Y}|})$, then $\vec{\varepsilon}_{\mathcal{Y}}^T \cdot \mathbf{f}_{\vec{\theta}_0}$ can be viewed as a weighted sum of each dimension of vector $\vec{\varepsilon}_{\mathcal{Y}}$, thus satisfying a one-dimension Gaussian distribution, i.e. $\vec{\varepsilon}_{\mathcal{Y}}^T \cdot \mathbf{f}_{\vec{\theta}_0} \sim \mathcal{N}(0, \frac{\mathbf{f}_{\vec{\theta}_0}^T \cdot \Sigma \cdot \mathbf{f}_{\vec{\theta}_0}}{|\mathcal{Y}|})$. Then $\Delta \mathbf{F}(\vec{\theta}_0)$ also satisfies a Gaussian distribution, i.e. $\Delta \mathbf{F}(\vec{\theta}_0) \sim \mathcal{N}(-\eta \mathbf{f}_{\vec{\theta}_0}^T \mathbf{f}_{\vec{\theta}_0}, \eta^2 \frac{\mathbf{f}_{\vec{\theta}_0}^T \cdot \Sigma \cdot \mathbf{f}_{\vec{\theta}_0}}{|\mathcal{Y}|})$. In practice, they use the approximate gradient $\hat{\mathbf{f}}_{\vec{\theta}_0}$ to calculate the mean and variance of $\Delta \mathbf{F}(\vec{\theta}_0)$.

For the covariance matrix of $\vec{\varepsilon}_{\mathcal{Y}}$, $\Sigma$, its unbiased estimation $\hat{\Sigma}$ is:

$$\hat{\Sigma} = \frac{\sum_{\mathbf{y}_j \in \mathcal{Y}} (\mathbf{g}_{\vec{\theta}_0}^j - \hat{\mathbf{f}}_{\vec{\theta}_0})(\mathbf{g}_{\vec{\theta}_0}^j - \hat{\mathbf{f}}_{\vec{\theta}_0})^T}{|\mathcal{Y}| - 1} \tag{4}$$

After modelling the decrease of objective value as a normally distributed random variable, Yin et al. (2017) abstract the process of objective value change as a random walk game with a Gaussian dice.

In the game, the objective value is regarded as a state. For the simplicity of notation, we define a state $s_t$ as below.

$$s_t = -\frac{\Delta \mathbf{F}}{\eta} \tag{5}$$

The decrease of objective value, namely the transfer from the current state to the next state is determined by a Gaussian dice, of which the mean solely depend on the current state and the variance is controlled by the state and the batch to be chosen. They define the domain of game state as a half closed set of real numbers $[S^*, +\infty)$, where $S^*$ is the minimum objective value that the learning can possibly achieve. The game starts with a random state and the goal is to move as close as possible to $S^*$. There are two ways of state transfer: one directly decreasing from $S_i$ to $S_j$, another first reaching minimum point $S^*$ and then rebounding to $S_j$.

Formally, for state $s_t$, the moving step $\triangle s_t$ is generated by a Gaussian dice $\mathcal{N}(\mu_t, \frac{\sigma_t^2}{m})$, where $\mu_t = \hat{\mathbf{f}}_{\vec{\theta}_t}^T \hat{\mathbf{f}}_{\vec{\theta}_t}$, $\sigma_t^2 = \hat{\mathbf{f}}_{\vec{\theta}_t}^T \hat{\Sigma} \hat{\mathbf{f}}_{\vec{\theta}_t}$, $m = |\mathcal{Y}|$, denoting the batch size. The state transition equation is presented below.

$$s_{t+1} = |s_t - S^* - \eta \triangle s_t| + S^* = \begin{cases} s_t - \eta \triangle s_t, & \text{if } s_t - \eta \triangle s_t \geq S^* \\ 2S^* + \eta \triangle s_t - s_t, & \text{otherwise} \end{cases} \tag{6}$$

Let $p_m^{s_t}(\triangle s_t)$ denote the probability density function for a random moving step $\triangle s_t \sim \mathcal{N}\left(\mu_t, \frac{\sigma_t^2}{m}\right)$. The expected value of next state can be expressed as follows.

$$\mathbf{E}_m^{s_t}(s_{t+1}) = \int_{-\infty}^{+\infty} p_m^{s_t}(\triangle s_t)(|s_t - S^* - \eta \cdot \triangle s_t| + S^*) d\triangle s_t$$

$$= (s_t - S^* - \eta \mu_t)\{\Phi(a) - \Phi(-a)\} + \frac{\eta \sigma_t}{\sqrt{m}} \sqrt{\frac{2}{\pi}} e^{-\frac{a^2}{2}} + S^* \tag{7}$$

$$\text{where } a = \frac{s_t - S^* - \eta \mu_t}{\eta \sigma_t} \sqrt{m}$$

To decide the best batch size, one should consider both the variance of estimated gradients and computation cost. Large batch can reduce the variance and therefore make more accurate updates, whereas it also requires more computations. Yin et al. (2017) then define a utility function to find a balance. Maximizing the utility means achieving the largest expected decrease of loss per data instance.

$$\mathtt{u}(m, s_t) = \frac{s_t - \mathbf{E}_m^{s_t}(s_{t+1})}{m} \tag{8}$$

$$m^* \leftarrow \arg\max_m \mathtt{u}(m, s_t) \tag{9}$$

where $m^*$ is the best batch size for the $(t+1)$-th iteration.

For more specifications on the BA-SGD algorithm, see Yin et al. (2017).

## 3 BATCH ADAPTIVE MOMENTUM

The main appeal of momentum is its ability to reduce oscillations and accelerate convergence (Goh, 2017). The idea behind momentum is that it accelerates learning along dimensions where gradient continues pointing to the same direction, and slows down those where sign of gradient constantly changes (Zeiler, 2012). Recent work on momentum, called YellowFin, an automatic tuner for both momentum and learning rate, helps momentum optimizer converge in even fewer iterations than ADAM on large ResNet and LSTM models (Zhang et al., 2017). Since it is both powerful and popular, we would like to also apply the batch adaptive framework to momentum optimizer.

### 3.1 MOMENTUM

Momentum utilizes past parameter updates with an exponential decay. Its way of update is given by

$$\vec{\theta}_{t+1} = \vec{\theta}_t - \mathtt{m}_t \tag{10}$$

$$\mathtt{m}_t = \rho \mathtt{m}_{t-1} + \eta \hat{\mathtt{f}}_{\vec{\theta}_t} \tag{11}$$

where $\mathtt{m}_t$ denotes momentum at the $t$-th iteration and $\rho$ is the rate controlling the decay of the previous parameter updates. Equation 11 can also be written in the following form.

$$\mathtt{m}_t = \eta \sum_{\tau=0}^{t} \rho^{t-\tau} \hat{\mathtt{f}}_{\vec{\theta}_\tau} \tag{12}$$

### 3.2 DERIVATION

Referring to Equation 3, we can analogically estimate the decrease of objective value in the following expression.

$$\Delta \mathtt{F}_t = \mathtt{F}(\vec{\theta}_{t+1}) - \mathtt{F}(\vec{\theta}_t) = \mathtt{f}_{\vec{\theta}_t}^T (\vec{\theta}_{t+1} - \vec{\theta}_t) = -\mathtt{f}_{\vec{\theta}_t}^T \mathtt{m}_t$$
$$= -\eta \mathtt{f}_{\vec{\theta}_t}^T \sum_{\tau=0}^{t} \rho^{t-\tau} \hat{\mathtt{f}}_{\vec{\theta}_\tau} = -\eta \mathtt{f}_{\vec{\theta}_t}^T \sum_{\tau=0}^{t} \rho^{t-\tau} (\mathtt{f}_{\vec{\theta}_\tau} + \vec{\varepsilon}_\tau) \tag{13}$$

where $\vec{\varepsilon}_\tau = \frac{\sum_{\mathbf{y}_j \in \mathcal{Y}_\tau} \vec{\xi}_j}{|\mathcal{Y}_\tau|}$ is the noise term which represents the difference between the real gradient and the estimated gradient calculated on a batch $\mathcal{Y}_\tau$ chosen at the $\tau$-th iteration. Though the estimated gradient from the previous iterations (i.e. $\tau = 0...t-1$) has noise, their batches which respectively determine their noise have already been selected, thus their noise is no longer a random variable but a constant. However, for the $t$-th iteration, we have not decided which batch to be sampled, therefore $\vec{\varepsilon}_t$ is indeed a random variable, and on the basis of CLT we know it is normally distributed, i.e. $\vec{\varepsilon}_t \sim \mathcal{N}(\mathbf{0}, \frac{\Sigma_t}{|\mathcal{Y}_t|})$. Based on this and the fact that real gradients, $\mathtt{f}_{\vec{\theta}_\tau}$ ($\tau = 0...t$), are all constants for the $t$-th iteration, we then have the decrease of objective value, $\Delta \mathtt{F}_t$, satisfying a one-dimensional Gaussian distribution, which is also experimentally verified in Appendix B.

We need to calculate the mean and variance of $\Delta \mathbf{F}_t$, but we prefer not to record all the $\hat{\mathbf{f}}_{\vec{\theta}_\tau}$ from previous iterations. Therefore, we construct a recurrence formula to avoid the trouble.

Let $\mathcal{P}_t = \sum_{\tau=0}^{t} \rho^{t-\tau}(\mathbf{f}_{\vec{\theta}_\tau} + \vec{\varepsilon}_\tau)$, then we have the following recurrence formula.

$$\mathcal{P}_t = (\mathbf{f}_{\vec{\theta}_t} + \vec{\varepsilon}_t) + \rho \mathcal{P}_{t-1} \tag{14}$$

Thus the mean and variance of $\mathcal{P}_t$ can be calculated in the following forms.

$$\mu_{\mathcal{P}_t} = \rho \mu_{\mathcal{P}_{t-1}} + \hat{\mathbf{f}}_{\vec{\theta}_t} \tag{15}$$

$$\Sigma_{\mathcal{P}_t} = \frac{\hat{\Sigma}_t}{|\mathcal{Y}_t|} \tag{16}$$

Now we have $\Delta \mathbf{F}_t \sim \mathcal{N}(-\eta \hat{\mathbf{f}}_{\vec{\theta}_t}^T \mu_{\mathcal{P}_t}, \eta^2 \hat{\mathbf{f}}_{\vec{\theta}_t}^T \Sigma_{\mathcal{P}_t} \hat{\mathbf{f}}_{\vec{\theta}_t})$. In the Gaussian walk game with rebound illustrated in Section 2, the Gaussian dice here satisfies $\triangle s_t \sim \mathcal{N}(\hat{\mathbf{f}}_{\vec{\theta}_t}^T \mu_{\mathcal{P}_t}, \hat{\mathbf{f}}_{\vec{\theta}_t}^T \Sigma_{\mathcal{P}_t} \hat{\mathbf{f}}_{\vec{\theta}_t})$.

For simplicity, we let $\sigma_t^2 = \hat{\mathbf{f}}_{\vec{\theta}_t}^T \hat{\Sigma}_t \hat{\mathbf{f}}_{\vec{\theta}_t}$, $\mu_t = \hat{\mathbf{f}}_{\vec{\theta}_t}^T \mu_{\mathcal{P}_t}$, $m = |\mathcal{Y}_t|$, thus $\triangle s_t \sim N(\mu_t, \frac{\sigma_t^2}{m})$. Then the expected value of the next state shares the same expression with Equation 7, though the mean of $\triangle s_t$ is different.

For batch adaptive momentum algorithm, we also adopt the utility function in Equation 8, and the best batch size is the one that maximizes the utility function, calculated through Equation 9.

### 3.3 ALGORITHM

At the end of this section, we would like to summarize how the batch adaptive momentum algorithm is implemented by presenting the pseudo code below. In the pseudo code, the $M$ stands for the total budget, i.e. the total number of instances used for training, the $m_0$ means sampling step, the smallest batch increment. The rest symbols have been introduced before. In this pseudo code, we aim to calculate the optimal batch size for each update step. When an optimal size is determined and it is larger than the current batch size, we need to add more instances to enlarge the batch. However, $s_t, \mu_t, \sigma_t$ will change every time we add more instances, leading to a different optimal size. Thus

---

**Algorithm 1** Batch-Adaptive Momentum

1: **procedure** BA-MOMENTUM$(\mathcal{X}, \vec{\theta}, \eta, \rho, M, m_0)$
2:     **while** $M > 0$ **do**
3:         $\mathcal{Y}_t \leftarrow \emptyset$
4:         **repeat**
5:             random sample $\mathcal{Z}$ from $\mathcal{X} - \mathcal{Y}_t$ with $|\mathcal{Z}| = m_0$
6:             $\mathcal{Y}_t \leftarrow \mathcal{Y}_t \bigcup \mathcal{Z}$
7:             calculate $\hat{\mathbf{f}}_{\vec{\theta}_t}, \hat{\Sigma}_t$ with $\mathcal{Y}_t$                 ▷ Equation (1) and (4)
8:             calculate $\mu_{\mathcal{P}_t}$ with $\mu_{\mathcal{P}_{t-1}}$ and $\hat{\mathbf{f}}_{\vec{\theta}}$                ▷ Equation (15)
9:             $s_t \leftarrow \mathbf{F}(\vec{\theta}|\mathcal{Y}_t), \mu_t \leftarrow \hat{\mathbf{f}}_{\vec{\theta}_t}^T \mu_{\mathcal{P}_t}, \sigma_t \leftarrow \sqrt{\hat{\mathbf{f}}_{\vec{\theta}}^T \hat{\Sigma}_t \hat{\mathbf{f}}_{\vec{\theta}}}$
10:             $m^* \leftarrow \arg\max_m \mathbf{u}(m, s_t)$ where $S^*$ is user-specified
11:         **until** $|\mathcal{Y}_t| \geq \min\{m^*, |\mathcal{X}|\}$
12:         $\mathbf{m}_t = \rho \mathbf{m}_{t-1} + \eta \hat{\mathbf{f}}_{\vec{\theta}_t}$
13:         $\vec{\theta} \leftarrow \vec{\theta} - \mathbf{m}_t$
14:         $M \leftarrow M - |\mathcal{Y}_t|$
15:         $t = t + 1$
16:     **end while**
17:     **return** $\vec{\theta}$
18: **end procedure**

---

in practice, we can only gradually increase the batch size until it becomes larger than or equal to a running estimate of the optimal batch size. Lastly, when implementing the algorithm, one may note that it is time-consuming to calculate the covariance matrix, $\hat{\Sigma}_t$. We will discuss a tradeoff and the computation cost of this algorithm in Appendix C.

## 4 EXPERIMENT

In this section, we present the learning efficiency of different learning algorithms on two deep learning models. One is the CNN for image classification on the dataset of Fashion MNIST (Xiao et al., 2017), and the other is a more complex RNN model for relation extraction on financial documents (Miwa & Bansal, 2016). To evaluate our learning efficiency regardless of the size of the data set, we use epoch as the unit of scanned data and one epoch means all instances in the whole training set. We calculate the training loss on the whole training set each time when the model has scanned one more epoch of data.

### 4.1 MODEL AND DATASET

Fashion MNIST is a data set of Zalando's article images, which consists of 60000 training samples and 10000 test samples. Each sample is a $28 \times 28$ grayscale image, associated with a label from 10 classes. We design a CNN model with 3 convolutional layers for this experiment.

Another model we use for relation extraction task is a bi-directional LSTM RNN (Gers, 2001). To train this model, we use 3855 training instances. A detailed description on the specific task and architecture of networks for the two experiments can be found in Appendix A.

### 4.2 LOWER LOSS WITH BUDGET LIMIT

In both experiments, 12 different optimization algorithms are used. They are BA-Momentum, BA-SGD, mini-batch momentum and mini-batch SGD with a fixed size of 32, 64, 128, 256 respectively, and manually adjusted mini-batch momentum and SGD (denoted as Manual-Momentum and Manual-SGD). Smith et al. (2017) proposed this manually adjusted mini-batch method that increases the batch size after every manually set number of epochs to reduce the noise scale of estimated gradients during training. Here we let "Manual" methods start with batch sizes of 32 and double their sizes every 25 epochs for image classification task (100 epochs as total budget) and every 75 epochs for relation extraction task (300 epochs as total budget), eventually their sizes will reach 256.

For simplicity, we denote mini-batch momentum with a fixed size of 32 as "Mini-32-Momentum". The same rule applies to "Mini-256-SGD" etc. We choose "Manual" methods and mini-batch methods with different sizes ranging from 32 to 256 as baselines. This is because with a budget limit, the small batch method can have more but less accurate updates, while the large batch method can make fewer but more accurate updates. More updates and accurate updates both can to some extent help the model achieve a lower loss, thus we use these small and large batch methods, together with "Manual" methods as baselines to make the comparative test with "BA" methods more convincing. For "BA" methods, the smallest batch increment, $m_0$, is 32.

The result of image classification task on Fashion MNIST is plotted in Figure 1 and Figure 3a. The observation shows that for six different momentum-based optimizations, BA-Momentum achieves a lower loss than the rest five methods after 100 epochs of training. Furthermore, BA-Momentum has fewer fluctuations in later training stage than most of the methods with fixed batch sizes. For the SGD-based optimizations, BA-SGD achieves a second lowest loss while the lowest one is achieved by Mini-32-SGD.

Now we take a look at the batch size change per iteration, plotted in Figure 3a. For both BA-Momentum and BA-SGD, the batch size keeps almost constant at 32 in an early stage, while it more frequently increases to larger sizes in later training stage. The tendency is especially evident for BA-SGD. The lines in Figure 3a for BA-SGD gets denser as iterations increase, indicating the batch size rises more frequently from 32 to larger sizes. Also, the largest possible batch size for BA-SGD enlarges from 64 to almost 500 as learning proceeds.

We display the result of relation extraction task in Figure 2 and Figure 3b. BA-Momentum still achieves the lowest loss. For SGD-based methods, BA-SGD has a similar curve with Mini-32-SGD, while BA-SGD fluctuates much less in later training stage and eventually reach a lower training loss than Mini-32-SGD does. The batch size change for BA-SGD on relation extraction task have the similar trend observed in the image classification result but much less frequent, and the batch size of BA-Momentum stays nearly constant at 32. This might be because each training instance in the relation extraction task is a sentence containing dozens of possible quadruples for binary

classification, and a batch of 32 instances is already a relatively large batch in terms of number of binary classifications. Furthermore, the test accuracies in Appendix D show that for this task the larger the batch size is, the worse the model will generalize, thus it is smart for BA-SGD and BA-Momentum to choose smaller sizes. In addition, one may note that there is a sudden rise of loss from 0.05 to 0.30 at the 224th epoch in Figure 2b during the training of BA-SGD and the loss soon drops back after this epoch. We presume it is due to encountering some very inaccurate estimates of gradients.

From Figure 3, we can see that BA-SGD tends to have much larger batch sizes than BA-Momentum does on both tasks. This might be because momentum-based method is better at reducing the uncertainty of updates by taking into account previous updates, thus it does not need that large a batch which SGD requires, to further reduce the uncertainty.

Above all, for the four figures below, the proposed "BA" methods reach the lowest loss in three cases and always perform better than the "Manual" methods. As for test accuracies, we present them in Appendix D.

Here we analyze why "BA" methods can achieve the lowest loss in most cases. The "BA" methods adopt a smaller batch in an early stage, which allows them to have more iterations per epoch, then they dynamically enlarge their batch size in later stage to reduce the noise of estimated gradients, because the noise has a larger impact on the accuracy of parameter update in later training stage. Therefore "BA" methods can conserve the fast decrease of Mini-32 methods in early stage and meanwhile keep decreasing rather than severely fluctuate like Mini-32 methods do in later stage.

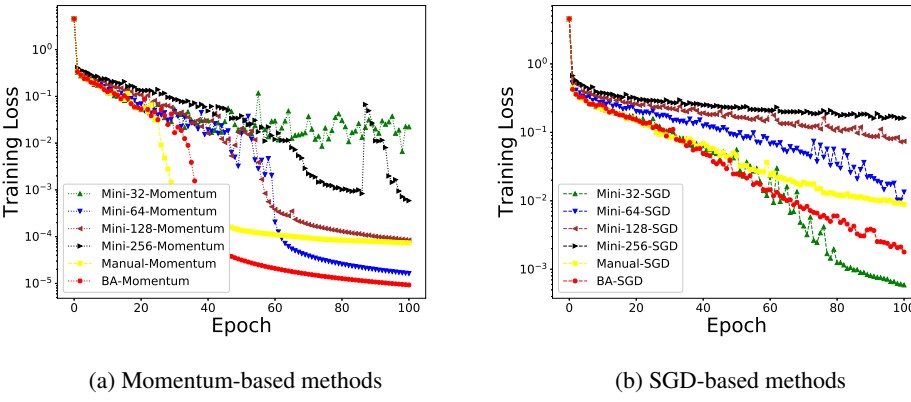

(a) Momentum-based methods

(b) SGD-based methods

Figure 1: Training loss per epoch on Fashion MNIST.

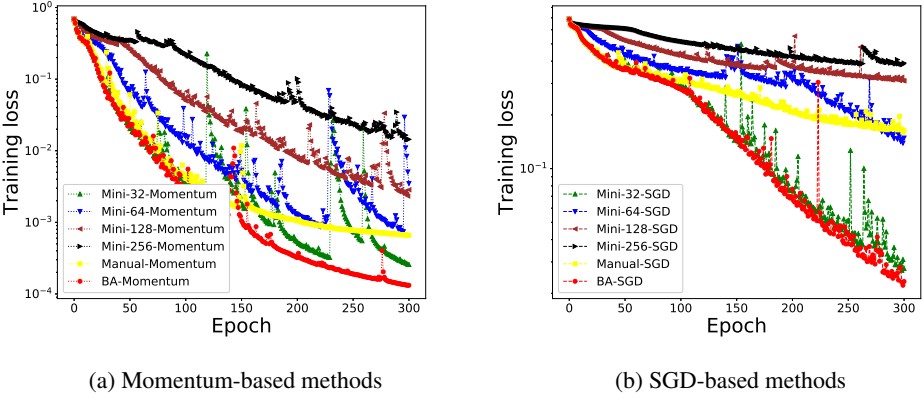

(a) Momentum-based methods

(b) SGD-based methods

Figure 2: Training loss per epoch on relation extraction.

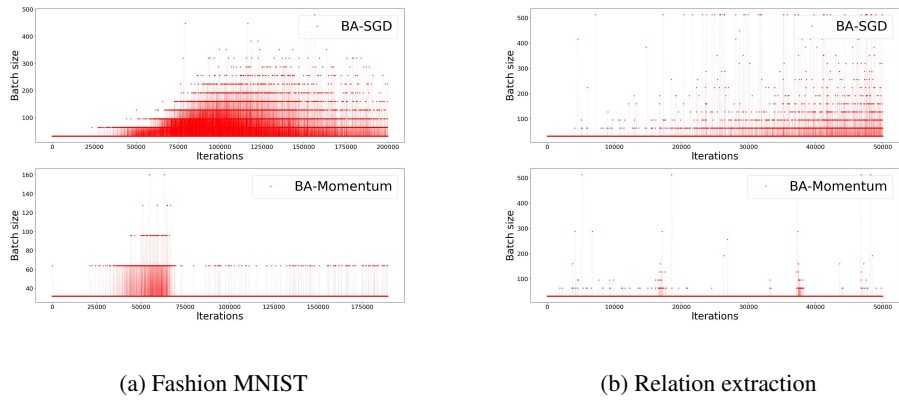

(a) Fashion MNIST          (b) Relation extraction

Figure 3: Batch size per iteration on Fashion MNIST and relation extraction.

## 4.3 ROBUSTNESS TO LARGE STEP SIZE

In the fine tuning process of BA-Momentum, we find that BA-Momentum is robust against large step sizes. Specifically, under a certain range of learning rates, when we tune the learning rate to a higher value, BA-Momentum can still achieve the same or even lower loss while the performance of other mini-batch methods will be degraded, converging to higher loss and fluctuating more. For simplicity, we only choose Mini-32-Momentum as baseline. We choose four different learning rates to test the robustness of BA-Momentum. They are 0.005, 0.01, 0.02, and 0.05.

The result is displayed in Figure 4. As shown in Figure 4a, when learning rate rises from 0.005 to 0.01 and then 0.02, BA-Momentum ends up with lower loss from 1.13e-4 to 1.69e-5 and then 5.77e-6. In contrast, Mini-32-Momentum fluctuates more when the learning rate rises from 0.005 to 0.01, and it ends up with much higher loss when the learning rate increases from 0.01 to 0.02. However, when the learning rate is tuned to 0.05, both BA-Momentum and Mini-32-Momentum suffer from a very high loss. These observations confirm that BA-Momentum is more robust against larger step sizes within a certain reasonable range.

We offer an explanation on the robustness of BA-Momentum. As the density of lines in Figure 4b suggests, the higher the learning rate is given, the larger our batch size tends to become. BA-Momentum enlarges the batch size to counteract with the larger uncertainty of parameter updates brought by higher learning rate. In this way, it successfully avoids the fluctuations and meanwhile benefits from the larger step size to converge faster.

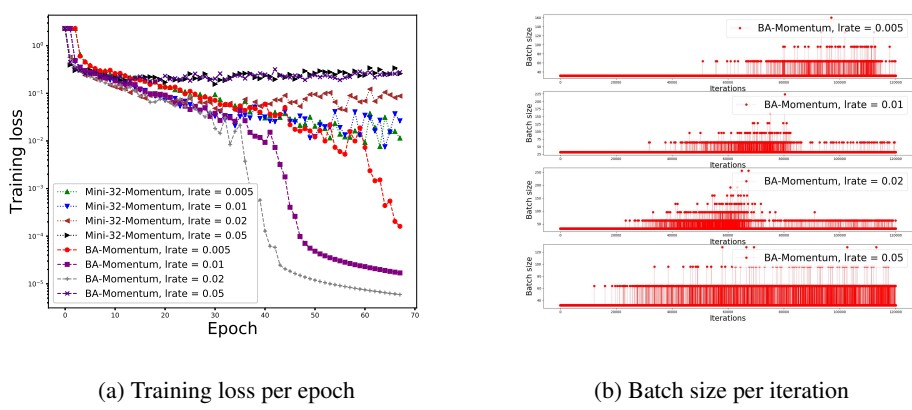

(a) Training loss per epoch          (b) Batch size per iteration

Figure 4: Effect of different learning rates on BA-Momentum.

### 4.4 BATCH SIZE BOOM

We find an intriguing phenomenon when conducting experiments testing the robustness of BA-Momentum. In the process of learning, the batch size of BA-Momentum will experience a *batch size boom*, a concentrated period where the batch size frequently increases to larger values. For example, the batch size boom in Figure 5a is from 80000 iterations to 110000 iterations, while the batch size boom in Figure 5b is from 60000 iterations to 80000 iterations. In the process of learning, when a batch size boom appears, the curve of BA-Momentum starts to diverge with the curve of Mini-32-Momentum. This makes sense because using a larger batch can help BA-Momentum make more accurate updates and thus decrease to a lower loss. Interestingly, the batch size boom always appears at the point in which Mini-32-Momentum reaches its lowest possible loss and after which it fluctuates around that loss. This can be observed in all three plots in Figure 5. As learning rate increases, Mini-32-Momentum reaches its lowest loss earlier, and the batch size boom also appears earlier, helping BA-Momentum keep decreasing to lower loss.

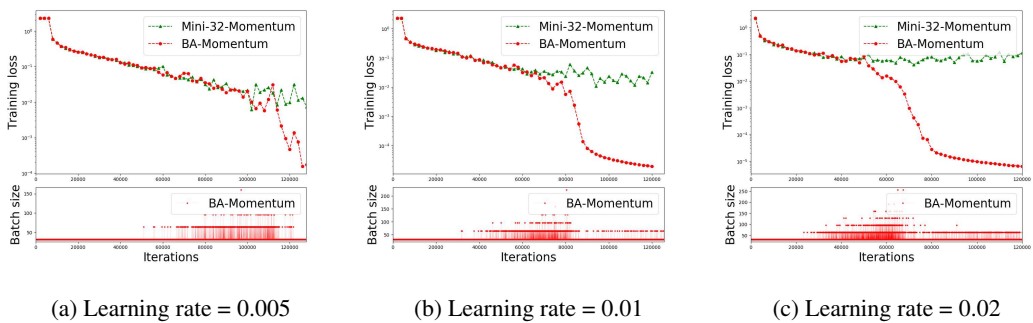

| (a) Learning rate = 0.005 | (b) Learning rate = 0.01 | (c) Learning rate = 0.02 |

Figure 5: Batch size boom on different learning rates.

## 5 CONCLUSION AND DISCUSSION

In this work we developed BA-Momentum algorithm, an extension of the BA-SGD proposed by Yin et al. (2017). We also evaluate the two algorithms on natural language processing and image classification tasks using RNN and CNN respectively. The experiments show that in most cases both batch adaptive methods can achieve lower loss than mini-batch methods after scanning same epochs of data. Furthermore, we also confirm that within a certain range of step sizes, BA-Momentum is more robust against large step size compared with mini-batch methods.

In the experiments, we did not evaluate the decrease of training loss with respect to training time. This is because, in the BA-SGD and BA-Momentum algorithm, we have to calculate the derivatives of the loss of each instance from a batch with respect to parameters, and then derive a covariance matrix through Equation 4 from the derivatives. Computing derivatives by backpropagation is time consuming, and now we have to compute all the derivatives of every instance in a batch. However, in mini-batch gradient descent, it is a common practice to calculate an average loss from a batch and then the derivative of this average loss, which requires less time. A feasible approach to reduce the computation cost might be to modify the way Theano do the derivation for a batch of instances and return the square sum of the derivatives, which we plan to study in future work.

The batch adaptive framework can have many important applications. It can be adapted to accelerate distributed deep learning. For distributed deep learning, communication cost for synchronizing gradients and parameters among workers and parameter server is its well-known bottleneck (Li et al., 2014a;b; Wen et al., 2017). A larger batch may help make more accurate updates, thus reducing the total number of iterations needed to converge, lowering the communication cost. However, a larger batch also causes a higher computation cost per iteration. In this update-costly environment, the batch adaptive framework may be modified to take both the computation and communication cost into consideration when deciding a proper batch size, which is worth further exploring.

Another application is that the batch adaptive framework may help remedy the generalization degradation of using large batch studied by Keskar et al. (2016). They provided solid numeric evidence

suggesting that using a larger batch will degrade the quality of the model, as measured by its ability to generalize. They also studied the cause for this generalization drop and presented evidence supporting the view that large-batch methods tend to converge to sharp minimizers of the training and testing functions, which causes this generalization drop. Several strategies to help large-batch methods eliminate this generalization drop was proposed in their work. The most promising one is to warm-start with certain epochs of small-batch regime, and then use large batch for the rest of the training. However, the number of epochs needed to warm start with small batch varies for different data sets, thus a batch adaptive method that can dynamically change the batch size against the characteristics of data is the key to solving this problem. The batch adaptive framework sheds light on this issue. Difficulty lies in how to identify a sharp minima accurately and efficiently in the process of learning and limit the batch size when encountering a sharp minima, which we plan to study in future work.

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

## A    ARCHITECTURE OF NETWORKS

For the classification task on Fashion MNIST, we use 3 convolutional layers, where each layer has 32, 64, 128 filters respectively, with MaxPool after each convolutional layer. For all layers, we use ReLU activations and Dropout (Srivastava et al., 2014).

The relation extraction task aims to extract quadruples in which the elements are correctly matched with each other. The input training instance consists of a sequence representing a sentence from financial documents, and all possible quadruples in this sequence. For the model, a bi-directional LSTM RNN is used. There are two layers of bi-directional LSTM, and a softmax outputs the probability of each quadruple being correct.

## B    MODEL VERIFICATION

In our BA-Momentum method, we model the decrease of objective value as a random variable satisfying a one-dimensional Gaussian distribution in Equation 13. This derivation is on the basis of first order Taylor extension and central limit theorem, involving some approximation. After modelling the decrease of objective value as normally distributed, we then estimated its mean and variance in Equation 15 and Equation 16. Here we would like to use real data from Fashion-MNIST to verify that the decrease of objective value truly satisfies a Gaussian distribution and our estimation is close to its real mean and variance.

We compute the decrease of objective value on the whole data set after 10 iterations for 500 times with batches randomly sampled and size fixed at 100. The result is plotted in Figure 6. The observation confirms that the decrease of objective value distributes normally. Its mean and standard deviation are -7.94253e-10 and 3.48414e-11 respectively, while its estimated mean and standard deviation are -8.58165e-10 and 4.58062e-12. Therefore, the decrease of objective value in our BA-M algorithm satisfies a Gaussian distribution and our estimation for its mean and variance is close to the real ones.

## C    IMPLEMENTATION ISSUES

When calculating the covariance matrix in Equation 4, one should note that it takes a space of $O(d^2)$ with $d$ denoting the dimension of the model parameter vector to store the covariance matrix. The cost is quite high for complex models with a large amount of parameters. Yin et al. (2017) has proposed a practical tradeoff assuming that model parameters are independent of each other, then the covariance matrix becomes a diagonal one, greatly reducing the space and time cost. Here we use a small data set from UCI machine learning repository (Lichman, 2013) to test whether the tradeoff will affect the performance of BA-Momentum.

We use two different BA-Momentum. One calculates the exact covariance matrix, while the other calculates its estimated diagonal matrix. The task is a classification task with 3 classes and 42 features. The training set is composed of around 60000 data instances. After training for 14 epochs, we yield the following result. The Figure 7a show the batch size change as learning proceeds for

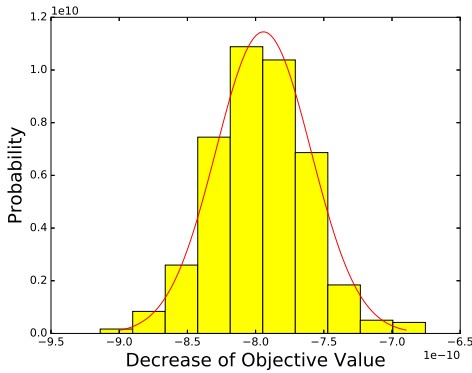

Figure 6: Verification on the distribution of decrease of objective value

matrix method and diagonal method respectively. The Figure 7b demonstrates different training loss per epoch for two different methods. From the figures we can see that the trend of batch size change for two methods are almost the same, therefore they perform equally well on the training loss descent. This indicates estimating the covariance matrix by its diagonal matrix will not cause degradation to the performance of BA-Momentum.

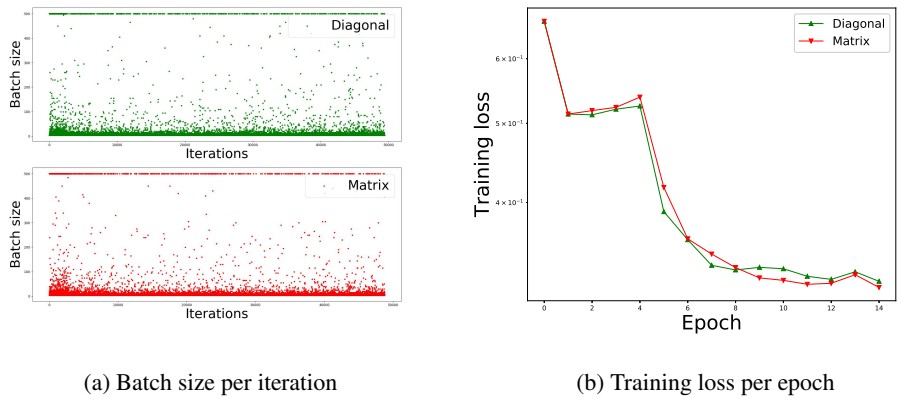

(a) Batch size per iteration          (b) Training loss per epoch

Figure 7: Effects of calculating exact covariance matrix and its estimated diagonal.

As for the cost of computing the optimal batch size, on average it takes up 1.03% and 0.61% of the total computing time per iteration for BA-Momentum and BA-SGD respectively on the image classification task, and the percentage on the relation extraction task is 1.31% and 0.92% for BA-Momentum and BA-SGD respectively. Computing the optimal batch size involves calculating some means and variances, and a binary search to find the $m^*$ that maximizes the utility function. Both operations take little time.

## D TEST ACCURACIES

We show the test accuracies of different methods on the image classification task and relation extraction task in Table 1. For the relation extraction task, we would like to mention that its accuracy is an intersection over union, specifically defined as the ratio of the intersection of the predicted quadruples and the ground truth quadruples to their union. For the evaluation, we record the test accuracies of each method after every epoch's training and present below the best test accuracies achieved by each method during their whole budget's training (100 epochs for image classification task and 300 epochs for relation extraction task).

The proposed "BA" methods achieve two best test accuracies in the following four cases (SGD-based on Fashion MNIST and SGD-based on relation extraction). In the other two cases, the test accuracies achieved by "BA" methods, 91.33% and 88.46%, are still very close to the best ones, 91.64% and 89.02% respectively. What's more, these results are realized in a self-adaptive way and require no fine tuning, while the best accuracies in the two momentum-based cases are achieved by totally different fixed batch sizes, indicating a tuning process. In contrast with "BA" methods, "Manual" methods which manually increase the batch size performs well on Fashion MNIST, whereas they ends up with the fourth highest test accuracies on the relation extraction task, indicating that "Manual" methods also need fine tuning to realize a satisfactory generalization.

Table 1: Best test accuracies achieved by different methods on Fashion MNIST and relation extraction

|          | Fashion MNIST |          | Relation extraction |          |
|----------|---------------|----------|---------------------|----------|
|          | Momentum      | SGD      | Momentum            | SGD      |
| Mini-32  | 90.93%        | 90.87%   | **89.02%**          | 75.13%   |
| Mini-64  | 91.17%        | 90.64%   | 87.12%              | 58.01%   |
| Mini-128 | **91.64%**    | 90.37%   | 85.33%              | 31.21%   |
| Mini-256 | 90.76%        | 90.16%   | 79.37%              | 28.11%   |
| Manual   | 91.35%        | 90.84%   | 86.31%              | 50.02%   |
| BA       | 91.33%        | **91.01%** | 88.46%            | **75.57%** |

