# OpenReview forum: "On Batch Adaptive Training for Deep Learning: Lower Loss and Larger Step Size"
_ICLR.cc/2018/Conference — Reject_

### Official Review · AnonReviewer1 · 2017-11-17
**Interesting and intuitive idea, but I'm not convinced that it adds enough over Yin et al. KDD paper, and think the experiments must include results on testing data.**

**Rating:** 5
**Confidence:** 3

**Review:**

The authors propose extending the recently-proposed adaptive batch-size approach of Yin et al. to an update that includes momentum, and perform more comprehensive experiments than in the Yin et al. paper validating their approach.

The basic idea makes a great deal of intuitive sense: inaccurate gradient estimates are fine in early iterations, when we're far from convergence, and accurate estimates are more valuable in later iterations, when we're close. Finding the optimal trade-off between computational cost and expected decrease seems like the most natural way to accomplish this, and this is precisely what they propose. That said, I'm not totally convinced by the derivation of sections 2 and 3: the Gaussian assumption is fine as a heuristic (and they don't really claim that it's anything else), but I don't feel that the proposed algorithm really rests on a solid theoretical foundation.

The extension to the momentum case (section 3) seems to be more-or-less straightforward, but I do have a question about equation 15: am I misreading this, or is it saying that the variance of the momentum update \mathcal{P} is the same as the variance of the most recent minibatch? Shouldn't it depend on the previous terms which are included in \mathcal{P}?

I'm also not convinced by the dependence on the "optimal" objective function value S^* in equation 6. In their algorithm, they take S^* to be zero, which is a good conservative choice for a nonnegative loss, but the fact that this quantity is present in the first place, as a user-specified parameter, makes me nervous, since even for a nonnegative loss, the optimum might be quite far from zero, and on a non-convex problem, the eventual local optimum at which we eventually settle down may be further still.

Also, the "Robbins 2007" reference should, I believe, be "Robbins and Monro, 1951".

These are all relatively minor issues, however. My main criticism is that the experiments only report results in terms of *training* loss. The use of adaptive batch sizes does indeed appear to result in faster convergence in terms of training loss, but the plots are in log scale (which I do think is the right way to present it), so the difference is smaller in reality than it appears visually. To determine whether this improvement in training performance is a *real* improvement, I think we need to see the performance (in terms of accuracy, not loss) on held-out data.

Finally, as the authors mention in the final paragraph of their conclusion, some recent work has indicated that large-batch methods may generalize worse than small-batch methods. They claim that, by using small batches early and large batches late, they may avoid this issue, and I don't necessarily disagree, but I think an argument could be made in the opposite direction: that since the proposed approach becomes a large-batch method in the later iterations, it may suffer from this problem. I think that this is worth exploring further, and, again, without results on testing data being presented, a reader can't make any determination about how well the proposed method generalizes, compared to fixed-size minibatches.

---

> ### Author Response · Authors · 2017-12-23
> **Response to AnonReviewer1**
>
> Thank you for your detailed comments on our work.
>
> You mentioned that the proposed algorithm in Page 5 might not rest on a solid theoretical foundation. We would like to make it clear that the algorithm is a trade-off in practice. In this algorithm, we aim to calculate the optimal batch size for each update step. When an optimal size is determined and it is larger than the current batch size, we need to add more instances to enlarge the batch. However, $s_t$, $\mu_t$, $\sigma_t$ will change every time we add more instances, leading to a different optimal size. Thus in practice, we can only gradually increase the batch size until it becomes larger than or equal to a running estimate of the optimal batch size.
>
> For the extension to momentum, the variance of the $\mathcal{P}_t$ is indeed the same as the variance of the most recent mini-batch, which is to be determined for the t-th iteration. This is because though the previous updates have their noises, their batches which respectively determined their noises have already been selected, thus their noises are no longer random variables but constants. This point has been clarified in the last paragraph in Page 4.
>
> As for S^*, thank you for your suggestion, it is indeed a user-specified parameter and the user can specify the S^* in terms of the specific definition of the loss function.
>
> Concerning the test accuracy, we would like to make it clear that the very aim of this batch adaptive method is to achieve the lowest loss possible within a certain budget of training data. It is the model’s aim, but not an optimizer’s aim to pursue a higher test accuracy. At your request, we still provide the test accuracy of each experiment, which can be found in Appendix D of the latest revised version. The results show that the proposed batch adaptive methods achieve two best test accuracies in all four cases (i.e. SGD-based and momentum-based for two tasks), and in the other two cases, the test accuracies achieved by batch adaptive methods, 91.33% and 88.46%, are still very close to the best ones, 91.64% and 89.02% respectively. What's more, these results are realized in a self-adaptive way and require no fine-tuning, while the best accuracies in the other two cases are achieved by totally different fixed batch sizes, indicating a tuning process.
>
> Finally, you expressed your concern about the feasibility of our method avoiding the generalization degradation of large-batch training. In fact, Keskar et al. (2016) studied the generation gap of large batch training and proposed one solution that is to warm-start with certain epochs of the small-batch regime, and then use large batch for the rest of the training. They examined this solution and it worked. However, the number of epochs needed to warm start with small batch varies for different data sets, thus a batch adaptive method that can dynamically change the batch size against the characteristics of data might be the key to solving this problem. Anyway, it is just a possibility worth exploring further.

---

### Official Review · AnonReviewer2 · 2017-11-22
**This manuscript addresses the problem of automatically tuning the batch size during deep learning training.  Experiments are conducted on CNN and LSTM RNN to demonstrate the advantages of the proposed method.**

**Rating:** 5
**Confidence:** 4

**Review:**

Overall, the manuscript is well organized and written with solid background knowledge and results to support the claim of the paper.  The authors borrow the idea from a previously published work and claim that their contributions are twofold: (1) extend batch adaptive SGD to adaptive momentum, and (2) adopt the algorithms to complex neural networks problems (while the previous paper only demonstrates with simple neural networks).  In this regard, it does not show much novelty.  Several issues should be addressed to improve the quality of the paper:
 1) The paper has demonstrated that the proposed method exhibits fast convergence and lower training loss.  However, the test accuracy is not shown.  This makes it hard to justify the effectiveness of the proposed method.
 2) From Fig. 4(b), it shows that the batch size is updated in every iteration.  The reviewer wonders whether it is too frequent.  Moreover, the paper does not explicitly show the computation cost of computing the batch size.
3) The comparison of other methodologies seems not fair.  All the compared methods adopt a fixed batch size, but the proposed method uses an adaptive batch size.  The paper can compare the proposed method with adaptive batch size in intuitive settings, e.g., small batch size in the beginning of training and larger batch size later.
4) The font size is too small in some figures, e.g., Figure 7(a).

---

> ### Author Response · Authors · 2017-12-23
> **Response to AnonReviewer2**
>
> We thank you for your constructive comments on our work.
>
> Concerning the test accuracy, we would like to make it clear that the very aim of this batch adaptive method is to achieve the lowest loss possible within a certain budget of training data. It is the model’s aim, but not an optimizer’s aim to pursue a higher test accuracy. At your request, we still provide the test accuracy of each experiment, which can be found in Appendix D of the latest revised version. The results show that the proposed batch adaptive methods achieve two best test accuracies in all four cases (i.e. SGD-based and momentum-based for two tasks), and in the other two cases, the test accuracies achieved by batch adaptive methods, 91.33% and 88.46%, are still very close to the best ones, 91.64% and 89.02% respectively. What's more, these results are realized in a self-adaptive way and require no fine-tuning, while the best accuracies in the other two cases are achieved by totally different fixed batch sizes, indicating a tuning process.
>
> As for the cost of computing the optimal batch size, on average it takes up 1.03% and 0.61% of the total computing time per iteration for BA-Momentum and BA-SGD respectively on the image classification task, and the percentage on the relation extraction task is 1.31% and 0.92% for BA-Momentum and BA-SGD respectively. Computing the optimal batch size involves calculating some means and variances, and a binary search to find the $m^*$ that maximizes the utility function. Both operations take little time.
>
> You suggest comparing the batch adaptive method with adaptive batch size in intuitive settings. We add one that doubles the batch size after certain epochs of training (see the revised version). This manually adjusted mini-batch method achieves a slightly higher test accuracy in one setting while it performs not so well in other three settings compared with our batch adaptive method. It demonstrates that this manual way of increasing batch size still requires manual setting to realize a satisfactory performance whereas our batch adaptive method is self-adaptive and it achieves a satisfactory test accuracy without fine-tuning.

---

### Official Review · AnonReviewer3 · 2017-11-27
**A small modification of adaptive batch size without momentum with unconvincing experiments**

**Rating:** 4
**Confidence:** 3

**Review:**

The paper proposes a generalization of an algorithm by Yin et al. (2017), which performs SGD with adaptive batch sizes. The present paper generalizes the algorithm to SGD with momentum. Since the original algorithm was already formulated with a general utility function, the proposed algorithm is similar in structure but replaces the utility function so that it takes momentum into account. Experiments on an image classification task show improvements in the training loss. However, no test accuracies are reported and the learning curves have suspicious artifacts, see below. Experiments on a relation extraction task show little improvement over SGD with momentum and constant batch size.


COMMENTS:

The paper discusses a relevant issue. While adaptive learning algorithms are popular in deep learning, most algorithms adapt the learning rate or the momentum coefficient, but not the batch size. It appears to me that the main idea and the overall structure of the proposed algorithm is the same as in the one published by Yin et al. (2017), and that only few changes were necessary to include momentum. Given the incremental process, I find the presentation unnecessarily involved, and experiments not convincing enough.

Concerning the presentation, the paper dedicates two full pages on a review of the algorithm by Yin et al. (2017). The first page of this review states that, for large enough batch sizes, the change of the objective function in SGD is normal distributed with a variance that is inversely proportional the batch size. It seems to me that this is a direct consequence of the central limit theorem. The derivation, however, is quite technical and introduces some quantities that are never used (e.g., $\vec{\xi}_j$ is never used individually, only the combined term $\epsilon_t$ defined below Eq. 12 is). The second page of the review seems to discuss the main part of the algorithm, but I could not follow it. First, a "state" $s_t$ (also written as $S$) is introduced, which, according to the text, is "the objective value", which was earlier denoted by $F$. Nevertheless, the change of $s_t$, Eq. 5, appears to obey a different probability distribution than the change of $F$. The paper provides a verbal explanation for this discrepancy, saying that it is possible that $S$ is first reduced to the minimum $S^*$ of the objective and then increased again. However, in my understanding, the minimum of the objective is only realized at a singular point in parameter space. Crossing this point in an update step should have zero probability as long as the model has more than one parameter. The explanation also does not make it clear why the argument should apply to $S$ (or $s$) but not to $F$.

Page 5 provides pseudocode for the proposed algorithm. However, I couldn't find an explanation of the code. The code suggests that, for each update step, one gradually increases the batch size until it becomes larger or equal than a running estimate of the optimal batch size. While this may be a plausible strategy in practice, it seems to have a bias that is not addressed in the paper: the algorithm recalculates a noisy estimate of the optimal batch size after each increase of the batch size, and it terminates as soon as the noisy estimate happens to be small enough, resulting in a bias towards a smaller than optimal batch size. A probably more important issue is that the algorithm is sequential and hard to parallelize, where parallelization is usually the main motivation to use larger batch sizes. As the gradient noise scales inversely proportional to the batch size, I don't see why increasing the batch size should be preferred over decreasing the learning rate unless optimizations with a larger batch size can be parallelized. The experiments don't compare the two alternatives.

Concerning the experiments, it seems peculiar that the learning curves in Figure 1 remain at a constant value for a long time at the beginning of the optimization before they begin to drop. Do the authors understand this behavior? It could indicate that the magnitude of the random initialization was chosen too small. I.e., the parameters might have been initialized too close to zero, where the loss is stationary due to symmetries. Also, absolute values of the training loss can be deceptive since there is often no natural scale. A better indicator of convergence would be the test accuracy. The identification of the "batch size boom" is interesting.

---

> ### Author Response · Authors · 2017-12-23
> **Response to AnonReviewer3**
>
> Thank you for the detailed comments.
>
> The key idea of the extension to momentum is that we need to consider the past parameter updates when calculating the mean of change of objective value, while its variance is only determined by the $\epsilon_t$, the noise at the t-th iteration. This is because, although the previous updates have their noises, their batches which respectively determined their noises have already been selected, thus their noises are no longer random variables but constants. In this sense, we argue that the extension to momentum might not be unnecessary.
>
> You asked about the definition of a “state” $s_t$ in the algorithm from Yin et al. (2017) in Page 3. Due to the page limit, this algorithm may lack more specific illustrations. Here a “state” $s_t$ is defined as $s_t = - \delta F / \eta$, where $\delta F$ is the change of objective value and $\eta$ is the learning rate. We omitted the definition of this denotation in our paper. Now it has been added in the latest revised version. We apologize for the confusion.
>
> As for the proposed algorithm in Page 5, it is indeed sequential. However, for parallelization, the largest possible batch size per iteration is limited by the computing resources. Thus our algorithm provides a way to examine whether it is the best time to update the parameters after a sequence of paralleled computations if we set the increment $m_0$ (a key parameter in the algorithm) to be the largest possible batch size that the computing resources allow.
>
> As you mentioned, we did not compare increasing the batch size with decreasing the learning rate. However, decreasing the learning rate is not an alternative to our batch adaptive method, it is complimentary. The batch adaptive method actually finds the optimal batch size adapted to different learning rates and different data sets. No matter how the learning rate is set, kept constant or decreasing, the algorithm still attempts to find the optimal batch size for each iteration adapted to the experiment’s settings, since it takes the learning rate into account when deciding the optimal batch size, see Eq. 7,8,9. The experiment in Section 4.3 also suggests the batch adaptive method can dynamically adapt the batch size against different settings of the learning rate within a certain range.
>
> You suggested that the magnitude of our random initialization is too small which causes the learning curves in Figure 1 and 2 remaining at constant values for several epochs at the start of the training, we scaled up the magnitude of random initialization, and then the curves drop much earlier than before, which can be seen in the latest revised version of the paper. We really appreciate your suggestion!
>
> Concerning the test accuracy, we would like to make it clear that the very aim of this batch adaptive method is to achieve the lowest loss possible within a certain budget of training data. It is the model’s aim, but not an optimizer’s aim to pursue a higher test accuracy. At your request, we still provide the test accuracy of each experiment, which can be found in Appendix D of the latest revised version. The results show that the proposed batch adaptive methods achieve two best test accuracies in all four cases (i.e. SGD-based and momentum-based for two tasks), and in the other two cases, the test accuracies achieved by batch adaptive methods, 91.33% and 88.46%, are still very close to the best ones, 91.64% and 89.02% respectively. What's more, these results are realized in a self-adaptive way and require no fine-tuning, while the best accuracies in the other two cases are achieved by totally different fixed batch sizes, indicating a tuning process.
>
> Lastly, thank you for your interest in “batch size boom”.

---

### Author Response · Authors · 2018-01-04
**We list the changes made in the revised paper**

We would like to bring to the reviewers’ attention that several changes have been made in the latest revised paper. They are listed below.
1. We add a comparison with the manually adjusted mini-batch method in Section 4.2, as one of the reviewers suggested. Figure 1, 2, 3 and the corresponding descriptions of results are all updated.
2. Test accuracies achieved by each method are presented in Appendix D.
3. We provide the cost of computing the optimal batch size in Appendix C.
4. We add more clarifications of the proposed pseudocode in Section 3.3.

---

### Decision · Program_Chairs · 2018-01-29
**ICLR 2018 Conference Acceptance Decision**

**Decision:**

Reject

**Comment:**

The reviewers generally thought the proposed algorithm was a straightforward extension of Yin et al., 2017, and not enough for a new paper.  They also objected to a lack of test results (to show generalization), but the authors did provide these in their revision.

Pros:
+ Adaptive batch sizing is useful, especially if the larger batches license parallelization.

Cons:
- Small, incremental change to the algorithm from Yin et al., 2017
- Test performance did not improve over well-tuned momentum optimization, which limits the appeal of the method.